# Physiological Significance of the Heterogeneous Distribution of Zeaxanthin and Lutein in the Retina of the Human Eye

**DOI:** 10.3390/ijms241310702

**Published:** 2023-06-27

**Authors:** Wojciech Grudzinski, Rafal Luchowski, Jan Ostrowski, Alicja Sęk, Maria Manuela Mendes Pinto, Renata Welc-Stanowska, Monika Zubik-Duda, Grzegorz Teresiński, Robert Rejdak, Wieslaw I. Gruszecki

**Affiliations:** 1Department of Biophysics, Institute of Physics, Maria Curie-Sklodowska University, 20-031 Lublin, Poland; rafal.luchowski@umcs.pl (R.L.); mmmendespinto@gmail.com (M.M.M.P.); monika.zubik@gmail.com (M.Z.-D.); 2Chair and Department of General and Pediatric Ophthalmology, Medical University of Lublin, 20-059 Lublin, Poland; jasion1@gmail.com (J.O.); robertrejdak@yahoo.com (R.R.); 3The National Institute of Horticultural Research, 96-100 Skierniewice, Poland; a.kazmierczuk@tlen.pl; 4Institute of Agrophysics, Polish Academy of Sciences, 20-290 Lublin, Poland; r.welc@ipan.lublin.pl; 5Chair of Forensic Medicine, Medical University of Lublin, 20-059 Lublin, Poland; grzegorz.teresinski@umlub.pl

**Keywords:** macula, retina, zeaxanthin, lutein, AMD

## Abstract

Zeaxanthin and lutein are xanthophyll pigments present in the human retina and particularly concentrated in its center referred to as the yellow spot (macula lutea). The fact that zeaxanthin, including its isomer *meso*-zeaxanthin, is concentrated in the central part of the retina, in contrast to lutein also present in the peripheral regions, raises questions about the possible physiological significance of such a heterogeneous distribution of macular xanthophylls. Here, we attempt to address this problem using resonance Raman spectroscopy and confocal imaging, with different laser lines selected to effectively distinguish the spectral contribution of lutein and zeaxanthin. Additionally, fluorescence lifetime imaging microscopy (FLIM) is used to solve the problem of xanthophyll localization in the axon membranes. The obtained results allow us to conclude that one of the key advantages of a particularly high concentration of zeaxanthin in the central part of the retina is the high efficiency of this pigment in the dynamic filtration of light with excessive intensity, potentially harmful for the photoreceptors.

## 1. Introduction

The human eye is an organ equipped with the ability of precise and color vision thanks to the densely packed cone photoreceptors located in the central part of the retina [1,2]. It turns out that in the optical path of photons incident on this area of the retina, there is a structure called the macula [1,3]. The entire human retina contains yellow xanthophyll pigments, lutein (Lut), zeaxanthin (Zea), and *meso*-zeaxanthin (*m*-Zea, see Appendix A for chemical structures), but their concentration in the macula is ~100 times higher compared to the peripheral regions of the retina [4,5]. Moreover, the fraction of Zea and *m*-Zea in the central retina exceeds the fraction of Lut (Zea + *m*-Zea/Lut ≈ 2), and the proportion is inversed in the peripheral region (Zea + *m*-Zea/Lut ≈ 0.3) [1,6]. This heterogeneous distribution of macular xanthophylls in the human retina raises questions about the particularly high concentration of yellow pigments in the central region and the specific role of Zea and *m*-Zea in the macula. It has been proposed that one of the possible advantages of the higher accumulation of Zea compared to Lut is related to the higher efficiency of neutralization of reactive oxygen species potentially dangerous to photoreceptor membranes [7]. On the other hand, the yellow, xanthophyll-rich layer of the retina is not located near the photoreceptors, but the macular pigments are preferentially in the outer plexiform layer [8,9]. This suggests that, in addition to the anti-oxidation function of macular xanthophylls, other modes of their physiological activity have to be considered. Another widely accepted physiological function of macular xanthophylls is filtering short-wavelength radiation [1,3,8,10]. A particularly high concentration of macular xanthophylls in the central part of the retina seems to create excellent conditions for this role. In general, the retina is protected from excessive light by a regulatory mechanism based on pupillary constriction, but this process operates in the diameter range of 7–2 mm, so it does not protect the most central part of the retina called the fovea [11]. Fortunately, the macula covers this area of the retina. The xanthophyll pigments present in the macula effectively absorb blue light thanks to the system of conjugated double bonds forming their molecular skeleton [12]. On the other hand, the spectroscopic properties of Lut and Zea are not fundamentally different, which raises the question of the possible physiological significance of the accumulation of both types of pigments in the retina and their heterogeneous distribution. In the present work, we address these issues by examining human retinal preparations with the application of confocal resonance Raman microscopy and confocal fluorescence lifetime imaging microscopy (FLIM).

## 2. Results and Discussion

Figure 1 presents the resonance Raman spectra recorded with the application of a Raman microscopic system from three regions of the same retina preparation, the foveola, fovea, and parafovea, significantly differing in the Zea:Lut ratio, with the application of two laser lines, 488 nm and 514 nm. In principle, both selected wavelengths can be applied to record resonance Raman spectra of both Lut and Zea [10,13,14]. On the other hand, the fact that 488 nm is in resonance with the 0-0 vibrational absorption band of Lut makes this wavelength particularly well suited to analyze this pigment [10,14,15]. Moreover, the fact that 514 nm is practically beyond the main absorption band of Lut but still within the light absorption spectral region of Zea and *m*-Zea makes this wavelength particularly well suited to analyze xanthophylls with longer conjugated double bond systems (see Appendix A for the xanthophyll absorption spectra) [10,14]. Owing to such circumstances, the resonance Raman spectra recorded in different retina regions, with the laser lines 488 nm and 514 nm, are substantially different since they show different proportions of Lut and zeaxanthins (Zea and *m*-Zea) depending on the distance from the center of the retina. This fact is reflected in the shift of the position of the ν_1_ Raman band representing the –C=C– stretching vibrations: the maximum at 1526 cm^−1^ in the case of Lut and 1521 cm^−1^ for zeaxanthins. A more detailed analysis of the ν_1_ band recorded with the application of both the laser lines shows different spectral widths in the spectra recorded from different retina regions. This effect is associated with a small but substantial contribution, in particular of Zea and *m*-Zea in the spectra recorded with 488 nm, manifested by a spectral broadening of the ν_1_ band (see the dashed lines in Figure 1). Importantly, the comparison of the concentration maps of Lut and Zea in the retina, developed on the basis of analyses of Raman scattering carried out with lasers of different emission wavelengths, being in different degrees of resonance with individual groups of pigments, leads to non-identical results (see Figure 2, Figure 3, Appendix A). This means that the contribution of Lut or Zea to the Raman scattering signal may be underrated or overrated depending on the actual resonance conditions for these pigments. Such an effect can be clearly seen from the comparison of the validation dependencies based on the Raman spectra of mixtures of Zea and Lut recorded with different lasers, regardless of whether they are based on the Gaussian deconvolution of the principal spectral band ν_1_ (Figure 4) or on the classical least square spectral fitting of the entire spectra of pure components (see Appendix A). According to the obtained calibration relationships (Figure 4 and Appendix A), the Zea fractions are overrepresented based on the Raman scattering spectral analysis, especially in the case of the 514 nm laser line being in resonance with xanthophylls with a longer conjugated double bond system. A Raman spectral broadening effect of the mixture of two macular xanthophylls can also be used to visualize a relative distribution of Zea with respect to Lut (Figure 5). As can be seen, the concentration of Zea relative to Lut is particularly high in the very central part of the macula (foveola) and the central part (fovea) compared to the parafovea region (the distance > 500 µm from the center of the retina). Despite the fact that Zea fractions are also overrepresented in such a case, very clearly Zea and *m*-Zea concentrations significantly exceed that of Lut in the central retina. A detailed map of the distribution of Lut and Zea in the retina can be obtained on the basis of the analysis of components of resonance Raman scattering spectra [9] (see also Appendix A). As can be seen, despite the much lower concentration of Lut in the foveola compared to Zea, the profiles of the distribution of both xanthophylls show similar characteristics. A confocal Raman microscopy makes it also possible to analyze individual xanthophyll distribution in the direction perpendicular to the plane of the retina. Such an analysis is presented in Figure 6. The optical cross-section analysis of Zea and Lut distribution in the human retina, based on the Raman confocal microscopy, well represents the fact that the macular xanthophyll fraction is preferentially localized in the outer plexiform layer of the foveola [8,9]. As can be seen from the analysis of the cross-section profiles, both Zea and Lut are present basically in the same regions and structures of the retina, and the main difference is associated with the significantly higher concentration of Zea, compared to Lut, in the foveola and fovea regions of the retina. Such a difference can be a direct consequence of the distribution and abundance of membrane receptors specific to the xanthophyll-binding proteins and complexes such as HDL [16], GSTP1 [17], StARD3 [18], or albumin [19], which can potentially act as pigment transporters to various cellular structures and nano-compartments of the retina. Owing to the fact that the spatial resolution of the Raman microscopic system is not high enough to analyze single-cell membranes, we chose to use fluorescence lifetime imaging microscopy (FLIM) to analyze macular xanthophylls in single axons [10]. The disadvantage of this approach is that it does not give the ability to distinguish between Lut and Zea, but the advantages are very high sensitivity and spatial resolution [10]. In contrast to our previous study, in which we used the FLIM technique to study the orientation and localization of xanthophyll molecules in the axons forming the outer plexiform layer and oriented in the retinal plane [10], the aim of our current work was to record FLIM images of axon cross-sections perpendicular to the retinal plane and located between the outer plexiform layer and the photoreceptors (see Figure 7). The shape and position on the wavelength scale of the fluorescence emission spectrum recorded from a single pixel in the image of a single axon (Appendix A) confirm that the detected signal comes from carotenoids characterized by a conjugated system of double bonds N = 10 and N = 11 [10]. The distribution of high-intensity fluorescence in the left and right sectors of the axonal section, compared with the low signal level in the upper and lower sectors, clearly shows the transmembrane localization and orientation vertical to the membrane plane of xanthophyll molecules [20]. Such a conclusion resulting from the photoselection effect is additionally confirmed using the analysis of fluorescence anisotropy distribution (see Figure 7) [20]. Interestingly, despite significant changes in the power of the scanning laser, the fluorescence distribution directly representing the xanthophyll orientation does not change. This impression, based on the visual analysis of the images, is confirmed by the quantification of the fraction of the short-lifetime component (see Figure 7) directly representing the fluorescence of the macular xanthophylls [10]. As shown in our previous work, xanthophylls present in the axons forming the outer plexiform layer change their *trans*-*cis* molecular configuration in response to the changes in light intensity, which is the direct cause of the reorientation of Lut and Zea molecules in membranes [10]. Such transformations have been shown to result in significant changes in the absorption of blue light passing toward the photoreceptors through the xanthophyll-rich layer [10]. This light-intensity-controlled mechanism has been attributed a photoprotective role and has been termed “molecular blinds” [10]. As can be inferred from the analysis of the images presented in Figure 7 (see also Figure 8), the mechanism of “molecular blinds” operating in the outer plexiform layer effectively regulates the intensity of light passing through this layer, thanks to which changes in the scanning light intensity do not affect the xanthophylls located in the axons oriented vertically in relation to the plane of the retina and located in its more outer regions (closer to the photoreceptors). This particular experimental result confirms the very high effectivity of the mechanism based on the dynamic regulation of light intensity passing through the xanthophyll-rich retina layers towards the photoreceptors. On the other hand, this observation does not help explain the potential physiological significance of the heterogeneous distribution of Zea and Lut in the retina [9] (see Figure 5). It has been proposed that one of the advantages of Zea accumulation is the higher photoprotective efficiency of this xanthophyll in lipid membranes compared to Lut [7]. From the perspective of the “molecular blinds” mechanism, it is important to consider the difference between Lut and Zea in a photo-isomerization and/or reorientation in the lipid membrane between the vertical and horizontal positions [10]. The results of the computational analysis of the molecular dynamics have shown that the *all*-trans Zea is more effective than the *all*-trans Lut in adopting a vertical, transmembrane orientation within a lipid bilayer [21,22]. In our opinion, such a difference makes Zea particularly well suited to play the role of a central photo-active element of light-switchable “molecular blinds” enabling central and precise vision at low light intensities and protecting the photoreceptors against photodamage at very high light.

## 3. Materials and Methods

### 3.1. Retina Preparation

Human eyes were collected and processed by the Eye Tissue Bank and the Department of Forensic Medicine of Medical University in Lublin in compliance with the Guidelines for Good Clinical Practice. Owing to the fact that the human retina is fragile and sensitive to degradation, and the retina of the living organism may differ from the donor retina, the donor eyeballs were collected and examined in the shortest possible time post-mortem, within 6 to 12 h, in several cases after corneas had been removed for transplantation. All the experiments were conducted in accordance with the approvals of the Bioethics Commission affiliated with the Medical University of Lublin (decision KE-0254/100/2018 and KE-0254/66/03/2022).

The retina samples were prepared from the eyeballs immediately after the section, under dim light conditions. The vitreous, choroid, sclera, and other structures were separated, leaving the intact retina with a yellow macula spot distinctly visible. The samples in PBS were then transferred to quartz slides adequate for fluorescence and Raman imaging and immediately subjected to examination. For Raman imaging, the retina samples were transferred to 80% glycerol solution, cooled down to −30 °C and scanned using a temperature-controlled stage THMS600 (Lincam, Redhill, UK) with an additional thermocouple to monitor the temperature directly in the sample.

### 3.2. Raman Imaging

Raman imaging was carried out using an inVia confocal Raman microscope (Renishaw, Kingswood, UK) with an argon laser (Stellar-REN, Modu-Laser™, Centerville, UT, USA) operating at 488 nm, equipped with a 50x long distance objective (Olympus Plan N NA = 0.35). Optical images of the human retina were obtained and elaborated with WiRE 5.5 software (Renishaw, Kingswood, UK). Based on the optical image from the area of approx. 10 × 10 mm, a 3 × 3 mm area was selected, covering the yellow spot and imaged with a spatial resolution of 5 µm (pixel size 5 × 5 µm). In the case of cross-sectional measurements along the Z axis, the spatial resolution was of the order of one micrometer (size of a single voxel was 1 × 1 × 1 µm). In the case of all measurements, microscopic images were recorded with a light power of 500 μW or attenuated to lower laser powers if necessary (indicated). At each point of the Raman image map, the spectra were recorded with about 1 cm^−1^ spectral resolution (2400 lines/mm grating) in the spectral region 800–1800 cm^−1^ using the EMCCD detection camera Newton 970 (Andor, Belfast, UK). During the measurements, the human eye retina specimen was cooled to a temperature of −30 °C in a cooling holder for microscopic measurements using Linkam THMS600 (Redhill, UK). Images were acquired with the use of the Renishaw WiRE 5.5 system in high-resolution mapping mode (HR maps). The acquisition time for a single spectrum was 0.05 s. All spectra were pre-processed by cosmic ray removal and baseline correction using WiRE 5.5 software from Renishaw (Kingswood, UK). Raman spectra from the retina samples were typically recorded from the three regions referred to as foveola (the central disk diameter ~0.35 mm), fovea (the central disk diameter ~1 mm), and parafovea (the central disk diameter ~2 mm), which can be determined by marked changes in the macular xanthophyll concentration profile.

### 3.3. Fluorescence Lifetime Imaging Microscopy (FLIM)

Time-resolved measurements were carried out on a MicroTime 200 optical system manufactured by PiqoQuant GmbH (Berlin, Germany) with a special sample temperature stabilizer (BioCell™, JPK, Berlin, Germany) set to 36.6 °C for retina imaging. Measurements were performed using a 470 nm pulsed laser light with variable repetition rate adjusting the desired illumination intensity. The time resolution of the fluorescence intensity decays was set to 16 ps. The excitation light was focused on the sample using an Olympus IX71 (Shinjuku City, Tokyo, Japan) inverted microscope and a 60× NA1.3 oil-immersed objective. Fluorescence observation was carried out in confocal mode using a 50 µm pinhole, a ZT 470RDC dichroic mirror (Chroma-AHF Analysentechnik, Tübingen, Germany), along with 470 Notch StopLine and 550/88 bandpass filters (both from Semrock). The fluorescence signal was split using a polarization cube and directed to two identical single-photon counting module detectors (Excelitas Technologies, Göttingen, Germany). Analysis of the results was performed using SymPhoTime v. 2.6 software (PiqoQuant GmbH, Berlin, Germany).

Emission spectra coming directly from the axon membranes were recorded using a spectrograph (Shamrock 163) connected via a separate optical path to a MicroTime 200 microscope system. The stationary emission signal was detected using a Newton EMCCD DU970P BUF camera (Andor Technology, Belfast, UK) cooled to −50 °C.

Fluorescence anisotropy *r* was calculated according to the formula
$$r=\frac{I_{∥}-GI_{⊥}}{I_{∥}+2GI_{⊥}}$$
where *I*_∥_ is the fluorescence intensity parallel (to the direction of polarization of excitation light), *I*_⊥_ is the perpendicular fluorescence intensity, and *G* is the instrumental correction factor determined in a separate measurement with long lived fluorophore.

## 4. Conclusions

In the current work, we used confocal microscopic Raman imaging in the plane of the retina and in the direction perpendicular to it for a detailed analysis of the heterogeneous distribution of macular pigments Zea and Lut. The results of the analyses show a similar location of both xanthophylls, but a particularly high concentration of Zea in relation to Lut in the most central part of the macula. Comparison of the Raman spectroscopy-based images with the results obtained by the FLIM technique and the literature data of molecular modeling leads to the conclusion that one of the significant advantages of such a diverse distribution of macular xanthophylls is a particular predisposition of the Zea to be involved in regulatory activities in response to changes in light intensity. The results of this study also lead to the conclusion that the regulation of light intensity passing through the xanthophyll-rich outer plexiform layer is so effective that it does not lead to photoreaction of the xanthophylls present in the deeper layers of the retina close to the photoreceptors.

## Figures and Tables

**Figure 1 ijms-24-10702-f001:**
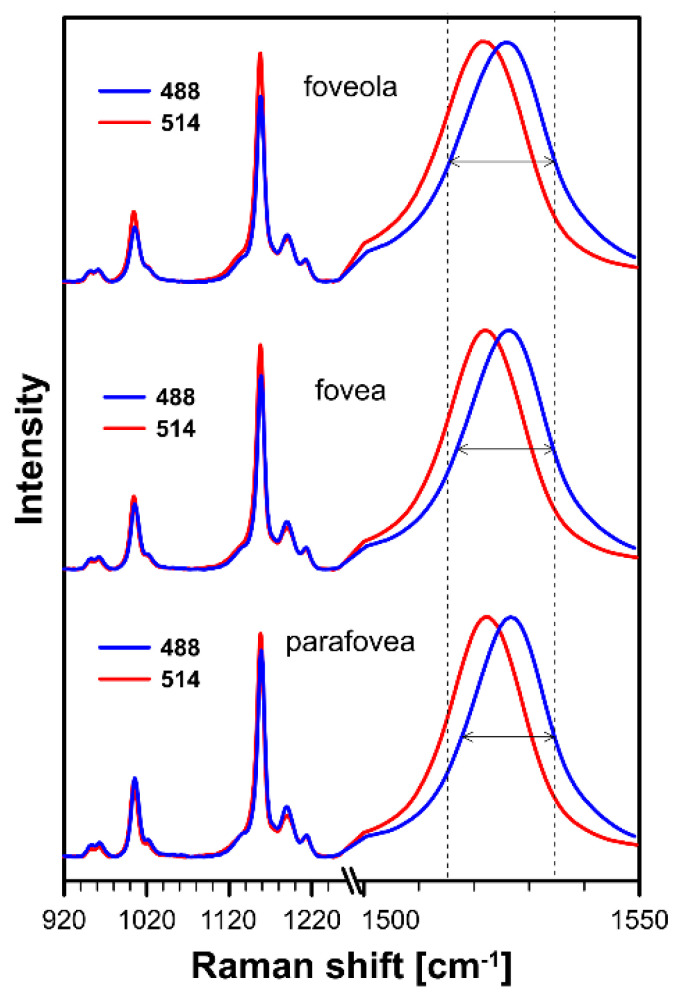
Raman spectra recorded with the application of a microscopic system from three different locations of preparation of the human retina: foveola, fovea, and parafovea. Spectra were recorded with two laser lines being preferentially in resonance with lutein (488 nm) or zeaxanthin and *meso*-zeaxanthin (514 nm). Note that the ν_1_ band representing the C=C stretching vibrations is shown on an enlarged scale. The retina is from a healthy 18-year-old female donor.

**Figure 2 ijms-24-10702-f002:**
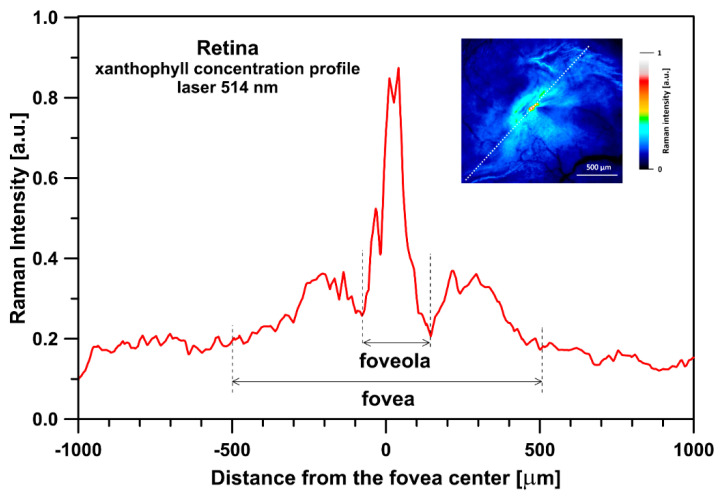
Optical cross-section analysis of the human retina based on the integration of the ν_1_ band of Raman spectra recorded with the application of a 514 nm laser line in the course of imaging the preparation shown in the inset. The direction of the cross-section is drawn with a dashed line. Detailed analysis of the contribution of lutein and zeaxanthin to the cross-section presented is shown in Appendix A. The retina is from a healthy 35-year-old male donor.

**Figure 3 ijms-24-10702-f003:**
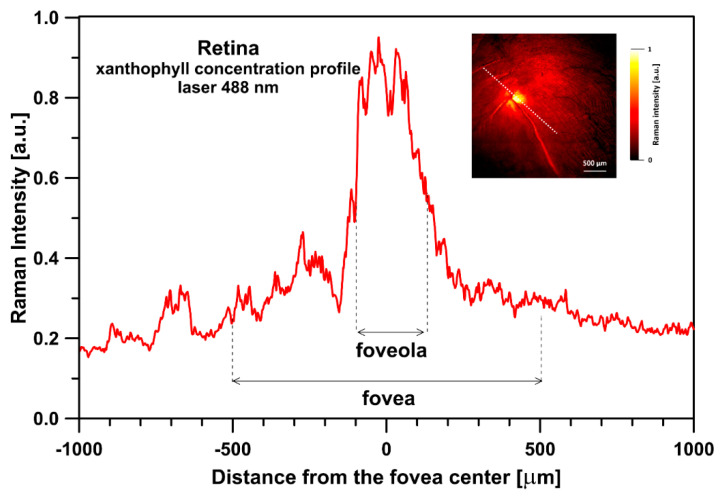
Optical cross-section analysis of the human retina based on the integration of the ν_1_ band of Raman spectra recorded with the application of a 488 nm laser line in the course of imaging the preparation shown in the inset. The direction of the cross-section is drawn with a dashed line. Detailed analysis of the contribution of lutein and zeaxanthin to the cross-section presented is shown in Appendix A. The retina is from a healthy 34-year-old female donor.

**Figure 4 ijms-24-10702-f004:**
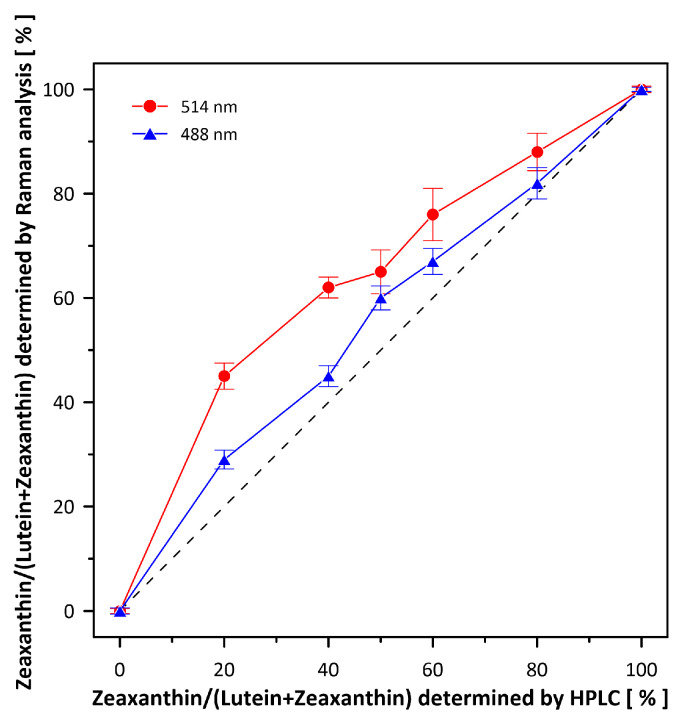
Validation dependencies of Raman spectroscopy-based determination of Lut and Zea fractions. Raman spectra were recorded from chromatographically pure 10-µM Lut and Zea mixtures in ethanol. Fractions of Lut and Zea were determined based on the Gaussian deconvolution of the ν1 spectral band (as shown in Appendix A). Note the deviations from the ideal linear relationship (dashed line) depending on the current resonance conditions for Lut and Zea scanned with the 488 nm and 514 nm lasers. Experimental points are the average from 10 different determinations ± S.D. R^2^ of the individual fits were in the range of 0.93 to 0.99.

**Figure 5 ijms-24-10702-f005:**
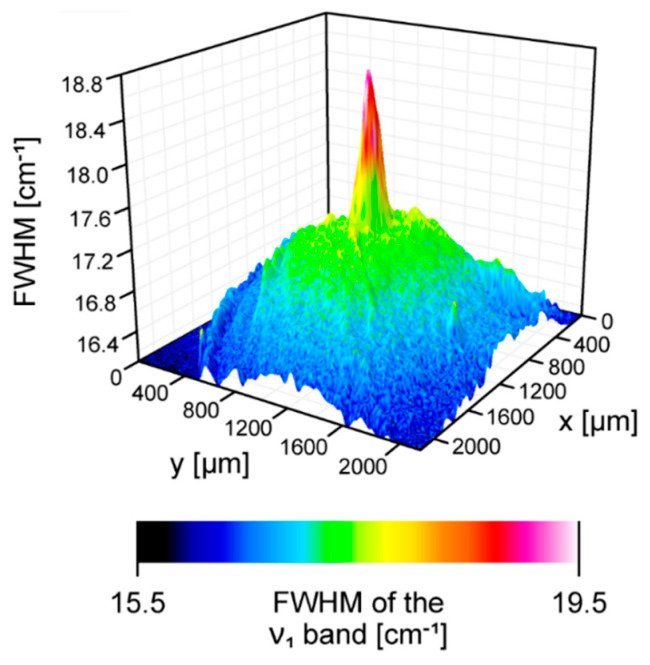
A 3D reconstruction of a concentration of zeaxanthin relative to lutein in the human retina sample imaged with the application of a Raman microscope. Spectra were recorded with a 488 nm laser line. The reconstruction is based on a full width at half maximum (FWHM) of the ν_1_ spectral band (see Figure 1). The retina is from a healthy 34-year-old female donor.

**Figure 6 ijms-24-10702-f006:**
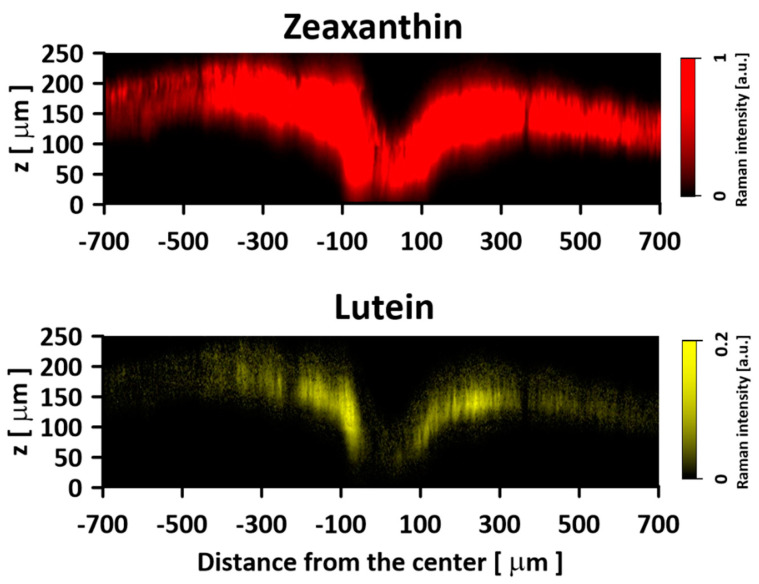
An optical cross-section of the human retina sample, showing the concentration distribution of zeaxanthin and lutein. Images are based on the Gaussian deconvolution of the ν_1_ band of the Raman spectra recorded in the course of Z-scan. Spectra were recorded with a 514 nm laser line. Note the different scales for the Lut and Zea distribution images. The retina is from a healthy 16-year-old male donor.

**Figure 7 ijms-24-10702-f007:**
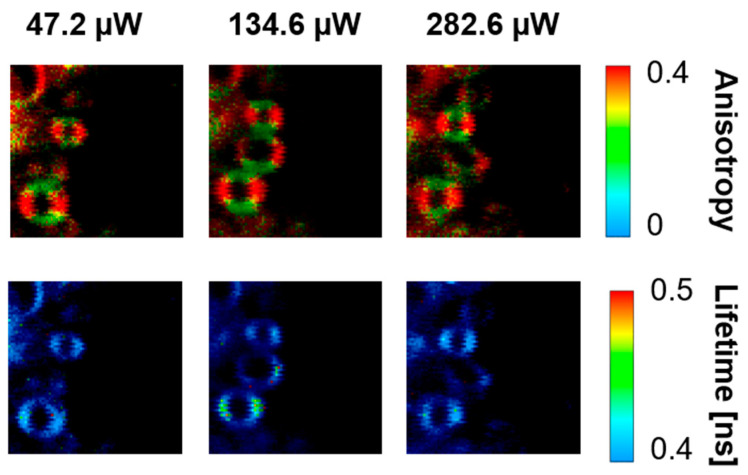
FLIM images of the optical cross-sections of single axons located between the outer plexiform layer and the photoreceptor layer of the human retina. The upper panels show fluorescence anisotropy, and the lower panels show fluorescence lifetime. Images were recorded with the different laser intensities shown above. The retina is from a healthy 57-year-old male donor.

**Figure 8 ijms-24-10702-f008:**
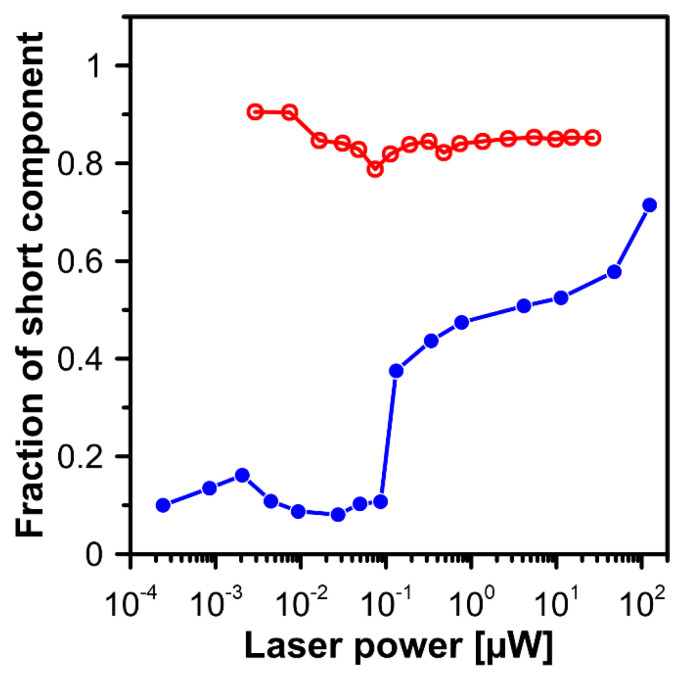
Laser power dependency of the fraction of the short-lifetime component in fluorescence recorded from axons located between the outer plexiform layer and the photoreceptor layer (images shown in Figure 6, red open symbols) and in the outer plexiform layer (the dependence redrawn from Figure 2c of the publication [10], blue filled symbols).

## Data Availability

Publicly archived datasets analyzed during the study can be found on the Internet: https://doi.org/10.5281/zenodo.7929829.

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
