# Peer review of "Physiological Significance of the Heterogeneous Distribution of Zeaxanthin and Lutein in the Retina of the Human Eye"

_ijms, 2023, doi:10.3390/ijms241310702_

Round 1

Reviewer 1 Report

The authors attempt to address the differential distribution of lutein and zeaxanthin in the human retina, which is an important problem in the macular carotenoid field. A similar study has been done with confocal resonance Raman microscopy and published with images at higher spatial resolution (ref. 9). In contrast, their Raman tests were done at -30 °C with both blue (488 nm) and green (514 nm) laser light on donor eyes aged from 19 to 57. Their main finding is that zeaxanthin is preferentially concentrated at the foveola, which confirms the findings of ref. 9 that used a 473 nm laser. This is an interesting result, obtained from the deconvolution of the Raman spectra of total macular carotenoids at 488 and 514 nm or by examination of FWHM of the v1 spectral band.  Since they are using different wavelengths and deconvolution methods, they should provide validation plots in their Supplementary Data section from prepared mixtures of lutein and zeaxanthin comparable to what was done in Figure 3 of ref. 9. 

Figure 6 is interesting in that it appears to show carotenoids in circular structures using FLIM, but it is confusingly written between the text and the legend.  In the text, it says they are measuring axons in the OPL, while the legend says “between the IPL and the photoreceptor layer”.  This is presumably the same, but they should use consistent terminology and provide a corresponding light microscopic image if available.  

The legend for Figure 7 is likewise confusing.  The text says their “molecular blinds” model of ref. 10 is for the xanthophylls in the OPL, but the blue data replotted in Figure 7 says it is for the IPL.  The red plot in Figure 7 from the OPL data in Figure 6 seems to be inconsistent with their model.  They should consider dropping this figure and shortening the corresponding text. 

Author Response

Answers to the points raised by the Reviewers

We would like to thank the Reviewer for the very constructive review and numerous comments and specific suggestions. The manuscript has been revised and the description of the changes made are presented below.

Reviewer #1

Reviewer comment:

The authors attempt to address the differential distribution of lutein and zeaxanthin in the human retina, which is an important problem in the macular carotenoid field. A similar study has been done with confocal resonance Raman microscopy and published with images at higher spatial resolution (ref. 9). In contrast, their Raman tests were done at -30 °C with both blue (488 nm) and green (514 nm) laser light on donor eyes aged from 19 to 57. Their main finding is that zeaxanthin is preferentially concentrated at the foveola, which confirms the findings of ref. 9 that used a 473 nm laser. This is an interesting result, obtained from the deconvolution of the Raman spectra of total macular carotenoids at 488 and 514 nm or by examination of FWHM of the v1 spectral band.  Since they are using different wavelengths and deconvolution methods, they should provide validation plots in their Supplementary Data section from prepared mixtures of lutein and zeaxanthin comparable to what was done in Figure 3 of ref. 9. 

Answer:

We thank the Reviewer for this comment and suggestion. We find it very important to our work. Appropriate validation relationships were measured for both lasers and plots were incorporated into the revised manuscript Supplementary data as new Figure S6 and discussed in the revised version of our work.

Reviewer comment:

Figure 6 is interesting in that it appears to show carotenoids in circular structures using FLIM, but it is confusingly written between the text and the legend.  In the text, it says they are measuring axons in the OPL, while the legend says “between the IPL and the photoreceptor layer”.  This is presumably the same, but they should use consistent terminology and provide a corresponding light microscopic image if available. 

Answer:

We thank the Reviewer for this comment. It should stand in the legend “between the outer plexiform layer and the photoreceptor layer”. The text has been corrected.

Reviewer comment:

The legend for Figure 7 is likewise confusing.  The text says their “molecular blinds” model of ref. 10 is for the xanthophylls in the OPL, but the blue data replotted in Figure 7 says it is for the IPL.  The red plot in Figure 7 from the OPL data in Figure 6 seems to be inconsistent with their model.  They should consider dropping this figure and shortening the corresponding text. 

Answer:

Unfortunately, in the caption to Figure 7, it was the same mistake as in the caption to Figure 6. It should stand “outer plexiform layer” instead of “inner plexiform layer”. The same as in the case of our previous publication (ref. 10). The text has been corrected and we are very thankful for pointing out this mistake.

Reviewer 2 Report

I have carefully read this manuscript. The authors used resonance Raman spectroscopy  to distinguish the spectral contribution of lutein and zeaxanthin in human retina. Also, FLIM was used to solve the problem of xanthophyll localization in axon membranes. They find that one of the key advantages of a high concentration of zeaxanthin in the central retina is the high efficiency of this pigment in the dynamic filtration of light with excessive intensity.

On the whole, this article has some novelty and scientific value, but there are still some problems, so I suggest the authors revise the article.

1. Some small language problems need to be polished;

2. I suggest the authors add the "Discussion (and conclusion)" part

3. The retina obtained after the donor's death is different from the living retina, and the authors need to mention this in the "Limitations" section

4. The author needs to clearly state "foveola",“"fovea"” and "parafovea" in the "Methods and Materials" section

Some small language problems need to be polished

Author Response

Answers to the points raised by the Reviewers

We would like to thank the Reviewer for the very constructive review and numerous comments and specific suggestions. The manuscript has been revised and the description of the changes made are presented below.

Reviewer #2

Reviewer comment:

I have carefully read this manuscript. The authors used resonance Raman spectroscopy  to distinguish the spectral contribution of lutein and zeaxanthin in human retina. Also, FLIM was used to solve the problem of xanthophyll localization in axon membranes. They find that one of the key advantages of a high concentration of zeaxanthin in the central retina is the high efficiency of this pigment in the dynamic filtration of light with excessive intensity.

On the whole, this article has some novelty and scientific value, but there are still some problems, so I suggest the authors revise the article.

  1. Some small language problems need to be polished;

Answer:

The language has been checked and corrected in several places.

Reviewer comment:

  1. I suggest the authors add the "Discussion (and conclusion)" part

Answer:

As suggested by the Reviewer, we expanded our discussion and added a Conclusions section.

Reviewer comment:

  1. The retina obtained after the donor's death is different from the living retina, and the authors need to mention this in the "Limitations" section

Answer:

This information has been incorporated into a revised version of our article.

Reviewer comment:

  1. The author needs to clearly state "foveola",“"fovea"” and "parafovea" in the "Methods and Materials" section

Answer:

This information has been incorporated into the Materials and Methods” section of a revised version of our article, as suggested by the Reviewer.

Round 2

Reviewer 1 Report

Overall, this version is substantially improved over the prior one.  

The validation of Raman determination of lutein and zeaxanthin in Figure S6 is very important, and I suggest that it be moved into the main text instead of in the supplementary data.  Their deviation from ideal values, especially when using the 514 nm laser is indeed concerning and is appropriately addressed in the text.  However, this figure could be improved because the components of lutein and zeaxanthin in Raman spectra were determined with CLS according to the legend, which is different from the Gaussian deconvolution algorithm mentioned elsewhere in the main manuscript and supplementary materials. Please be consistent. In addition, comparison analysis should be done properly. Typically, r2 and p values would be provided.

I am also concerned about their overemphasis of their "molecular blinds" hypothesis, especially in their concluding paragraph, as that is a minor focus of the experiments of the present manuscript.  As far as I am aware, cis-zeaxanthin has never been detected by HPLC in any human macular tissues under any lighting conditions in sufficient quantities to support their model.  The authors should acknowledge this shortcoming to provide a balanced portrayal of their current status of their hypothesis. 

Author Response

Answers to the points raised by the Reviewers

We thank the Reviewer #1 for further comments and specific suggestions. We consider them important to our work and have been guided by them in the preparation of the revised version of our manuscript.

Reviewer #1

Reviewer comment:

The validation of Raman determination of lutein and zeaxanthin in Figure S6 is very important, and I suggest that it be moved into the main text instead of in the supplementary data.  Their deviation from ideal values, especially when using the 514 nm laser is indeed concerning and is appropriately addressed in the text.  However, this figure could be improved because the components of lutein and zeaxanthin in Raman spectra were determined with CLS according to the legend, which is different from the Gaussian deconvolution algorithm mentioned elsewhere in the main manuscript and supplementary materials. Please be consistent. In addition, comparison analysis should be done properly. Typically, r2 and p values would be provided.

Answer:

We thank the Reviewer for this suggestion. We fully agree that showing a validation dependency in the main text will improve our work. We have prepared new validation dependencies based on the Gaussian deconvolution of the niu1 band of the Raman spectra of Lut and Zea mixtures and present them as a new Figure 4. We have also retained Figure S6 in order to have a comparison with the literature and other validation method.

Reviewer comment:

I am also concerned about their overemphasis of their "molecular blinds" hypothesis, especially in their concluding paragraph, as that is a minor focus of the experiments of the present manuscript.  As far as I am aware, cis-zeaxanthin has never been detected by HPLC in any human macular tissues under any lighting conditions in sufficient quantities to support their model.  The authors should acknowledge this shortcoming to provide a balanced portrayal of their current status of their hypothesis. 

Answer:

As we have shown in our previous publication (Luchowski et al., 2021), the cis forms of zeaxanthin (mainly 13-cis and 9-cis) have been detected by HPLC analysis of the human retina and their concentration has been shown to depend on the illumination (Figure 5 in this publication). In most of the previous studies, both ours and other laboratories, the HPLC columns used were unable to distinguish the geometric isomers of xanthophylls.

As suggested by the Reviewer we modified the Conclusion paragraph to stress the results shown in the present work.

Reviewer 2 Report

Can be accepted

Minor English editing needed

Author Response

We thank The Reviewer for the comments.